# Fatigue Crack Monitoring of T-Type Joints in Steel Offshore Oil and Gas Jacket Platform

**DOI:** 10.3390/s21093294

**Published:** 2021-05-10

**Authors:** Liaqat Ali, Sikandar Khan, Salem Bashmal, Naveed Iqbal, Weishun Dai, Yong Bai

**Affiliations:** 1College of Civil Engineering & Architecture, Zhejiang University, Hangzhou 310058, China; drali169@zju.edu.cn (L.A.); weishundai@zju.edu.cn (W.D.); baiyong@zju.edu.cn (Y.B.); 2Department of Mechanical Engineering, King Fahd University of Petroleum and Minerals, Dhahran 31261, Saudi Arabia; sikandarkhan@kfupm.edu.sa; 3Department of Electrical Engineering, King Fahd University of Petroleum and Minerals, Dhahran 31261, Saudi Arabia; naveediqbal@kfupm.edu.sa

**Keywords:** fatigue crack, damage detection, welded metallic joints, finite element method, structural health monitoring, piezoelectric sensors

## Abstract

Several approaches have been used in the past to predict fatigue crack growth rates in T-joints of the offshore structures, but there are relatively few cases of applying structural health monitoring during the non-destructive testing of jacket platforms. This paper presents an experimental method based on the sensing of the piezoelectric sensors and finite element analysis method for studying the fatigue cracks in the offshore steel jacket structure. Three types of joints are selected in the current research work: T-type plate, T-type tube-plate, and T-type tube joints. The finite element analysis model established in the current study computes and analyzes the high stress and high strain regions in the T-type joints. The fatigue damage in the T-type joints was successfully detected by utilizing both the finite element analysis and experimental methods. The results showed that fatigue cracks of the three types of joints are prone to appear at the weld toe and spread in the welding direction. The fatigue damage location of T-type plate and T-type tube-plate joints is more concentrated in the upper weld toe area, and the fatigue damage location of the T-type tube joint is closer to the lower weld toe area.

## 1. Introduction

In the last few decades, a number of jacket structures were built to mine subsea resources and many of them have already been operated for over 20 years. Therefore, technical assessments, such as environmental condition analysis and corrosion or fatigue analysis, are needed to continuously check the condition of these jacket structures. Structural reliability of the offshore structures has been discussed in the literature by Madsen et al. [1], Moan [2], and Melcher [3]. Offshore pipelines are used to carry oil and gas from wellhead to manufacture services. In 2016, more than 2000 metric tons of oil and gas leaked from these offshore facilities. The most common failure mode in offshore structures is the one caused by the fatigue [4,5]. The structure may also fail when the load is more than the yield strength [6]. The steel jackets structure has been extensively used in the offshore oil and gas facilities for the past two decades [7] in (Bohai Bay) China, Gulf of Mexico, and Brazil. When the structure reaches the design service, the jacket plays a most important role in structural aging when exposed to environmental circumstances such as waves currents and wind loads. The final strength deteriorates with the progress of cracks as well as corrosion, producing the risk of collapse or failure of the entire jacket and causes losses of life, property and possibly causes environmental contamination. Few features of aged jacket fatigue life re-evaluation are given in [8], where the failure evaluation was achieved by the pushover technique. The key purpose for this re-evaluation might be the need to extend safety and life expectancy.

Studies on the worldwide nonlinear collapse analysis of three-dimensional steel jackets have also been performed in the literature [9]. The strength of the tubular structure as a ring-cured DT fitting in offshore jackets was analyzed using a finite element technique by Lee and Parry [10]. Reliability fatigue analysis was performed on jacket support considering corrosion as well as inspection by Dong et al. [11]. The effects of corrosion and cracks on the ultimate strength of aging ships were studied by Paik and Thayamballi [12]. Reliability analysis of damaged ships was also conducted where the resistance of older ships was reduced to initial yielding, which would underestimate or miscalculate the final strength [13,14]. In the literature, the collapse of the ship’s plate due to cracks, dents, and corrosion was evaluated by experimental and numerical approaches [15,16]. Based on experimental and numerical outcomes, an overall expression of the final strength of the plate with cracks was obtained by Paik and Kumar [17]. A recent experimental survey of tiny box beams with mild, average, and severe corrosion was conducted by Saad et al. [18,19].

Various studies in the literature are focused on loading, operation, structural harm and on the decay of the fixed and floating offshore structures [20,21,22]. Because of the complication of the subject and all its parameters, it is impractical to collect an overall fatigue analysis of all relevant variables. As a result, some researchers have concentrated on the individual significance of a topic to global panoramas. Jimenez-Martinez [23], reviewed the statistical fatigue damage assessment of offshore fatigue under random loading. It was concluded that experimental evaluations must describe the scattering of structural and macro layers, including substances from environmental conditions and random reactions.

Velarde et al. [24,25,26] described the effects of the uncertainty of loading on inducing cumulative damage from fatigue. Zhu et al. [27] investigated the accumulation of nonlinear damage based on the heterogeneous curve model. Farhan et al. [28] recommended a fatigue life for an onshore hybrid wind turbine, based on the multi-axis damage criteria. Sabakhty and Khansari [29], calculated the dynamic reaction of jacket structure under linear also nonlinear wave models to investigate the validity of linear wave models in different ocean states and also developed a finite element model to estimate the fatigue life. Meng et al. [30] puts forward a reliability-based optimization scheme based on saddle point approximation, which is suitable for offshore structures. 

Zhang et al. [31] examined the effects of coupled wind loads on fixed tripod offshore wind turbines and floating fatigue assessments. Similarly, Fan et al. [32] evaluated cumulative fatigue damage of the offshore wind turbine structures with various wind loading conditions. Many other scientific studies in the literature have been performed related to the offshore jacket-type, fixed jacket platform, and offshore structural aspects [33,34,35,36,37], structural durability [38,39], new design solutions [40], nonlinear effects of fatigue analysis [41], time domain analysis gap effect [42], effects of stress concentration factors on offshore tubular connections [43,44], and S-N curves [45,46,47,48].

Fatigue is one of the main failure forms in offshore steel jacket platforms. To assess fatigue damage, these platforms are regularly inspected over their life span. Fatigue damage inspection information mainly includes crack measurements [49]. Moreover, the fatigue reliability evaluation of welding joints is carried out using two different methods, the first is based on stress life curve, also called linear damage accumulation law, the second is based on the fatigue crack growth analysis, as well as failure standards [49,50]. The first method is used during the design phase, to evaluate the reliability of the existing or planned design structure solutions [51]. The first method cannot contain information about the size of the crack while the second method relies on linear elastic fracture dynamics and can be used to check the reliability of the structures [52,53,54].

The birth of intelligent materials for example piezoelectric (PZT) sensors that are based on the non-destructive testing approach, has revolutionized the field of structural health monitoring. Although a comparatively new non-destructive test technique called electromechanical (EMI) technology has been studied for more than 20 years, there are still various problems that need to be solved before it can be applied to actual structures [55]. The well-designed structural health monitoring system minimizes the overall maintenance costs of the structure, by detecting the damage at an early stage, resulting in early action to prevent further damage. Piezoelectric transducer can function both as a sensor as well as actuator [56].

High-strength low-alloy (HSLA) steel was initially developed in the 1960s for offshore pipeline. Pipes are normally made of high-strength low-alloy steel because it is stronger and tougher than mild carbon steel [57,58]. Furthermore, HSLA steel are used in engineering construction [59], automotive [60], as well as offshore structures [61]. The main advantages of HSLA steel are that it reduces the thickness and weight of components and also reduces the welding costs [62,63,64,65]. There are three key techniques of underwater welding, local dry cavity, wet welding and drying. The most commonly used form is the wet welding in which arcing and joints are in direct contact with water [66,67,68,69,70]. Wet welding can be done by using shielded electrodes, flux leather wires and also using few other welding procedures [71,72,73,74,75]. 

As a welding environment, water causes many problems in the quality of the acquired joints. The most common problem in underwater welding is the cold cracking [76,77]. High cooling rates in wet welding results in emulsified structures. Welding joint structures have the characteristics of high hardness and low plasticity. Cold cracks are also called hydrogen cracks and it account for more than 90 percent of the actual steel structure welds cracks [78]. Cold welding usually occurs shortly after welding, generally within 48 hours. Due to high temperatures, water is converted into hydrogen (H_2_) gas in an arc. Hydrogen cracks are usually very fine and are hard to detect [79]. 

Practically, pre-heating is the extremely valuable way to avoid cold cracking [79]. The cold cracking behaviour is controlled by the thickness of the plate, the hydrogen content of the welding metal, the heat input during the welding process, the residual stress state of the welding area and also by the chemical composition of the metal [78,79]. It is vital to notice the relationship between welded metal microstructures and welding conditions. To this end, the scientific researchers pays attention to the susceptibility of steel to cold cracking by controlling welding parameters, preheating, applying low hydrological consumables and welding processes [80].

The form drilling and form tapping techniques can be used for the rapid and economical production of nutless bolted joins. Using these techniques, threaded holes can easily be produced on couples of dissimilar metal alloys, as it is the case of steels and aluminium alloys. During this technique a fastener can be introduced and screwed for achieving a tight bolted joint, without any necessity of nut, after the simultaneous form drilling on the aluminium–steel pairs. The form drilling and threading are performed on the same machine tool that reduces the whole process time. The target markets for this approach are the light boiler making industry, in order to eliminate either welding beads or classical bolted joints using nuts [81].

While performing the experimental procedure, it should be noted that it is very difficult to calculate some of the material properties through experiments due to the lake of adequate laboratories and specialists to conduct those challenging experiments. It is difficult to obtain some material parameters of the model through experiments. Many organizations are now able to afford erstwhile privileged equipment like micro-hardness tester, scanning electron microscope, atomic force microscopy, etc. However, the development in the computational front will help to reduce the dependence on the experiment and will make it possible to simulate many experiments from the techniques like crystal plasticity finite element method [82]. Silva et al., 2014 [83], developed innovative testing machines and methodologies for the characterization of materials. Their work was mainly focused on the design, fabrication, and instrumentation of a flexible drop weight testing machine, an electromagnetic compressive Hopkinson bar, and an electromagnetic cam-driven compression testing machine that are capable of performing the mechanical characterization of materials under medium and high rates of loading. It was shown that the newly proposed electromagnetic cam-driven compression testing machine equipped with a root type cam profile can successfully replicate the operating conditions of the two other testing machines.

Alternative processes to conventional machining are increasing in various fields. The conventional material removal processes excessively affects the surface integrity and functional characteristics related to surface state, such as tensile strength. The use of un-conventional machining processes like sheet metal forming and boiler making, abrasive waterjet AWJ, wire electro-discharge machining WEDM, laser and punching, could be used instead of the standardized milling process in order to obtain tensile specimens. There are two main advantages of using the alternative processes. The first advantage is that it shortens the time needed to produce tensile test specimens, and the second advantage is that the same technology finally applied in the final manufacturing step is used to prepare testing specimens [84]. During the process of carbon dioxide capture, storage and utilization, CO_2_ is transported by means of pipelines, with a variety of environmental conditions. The fatigue failure may be one of the possible causes of the pipeline’s failure and should be taken into account during the long term planning of these processes [85,86,87,88,89,90]. The offshore pipelines can failed by a variety of failure mechanisms that include fatigue failure, corrosion failure, internal sheath damage etc. The enhanced Risk Based Inspection (RBI) methodology is an efficient way of certifying the integrity of steel pipeline [91,92,93,94].

Offshore jacket platform is most important in the oil and gas exploitation area, which provides production and living facilities for offshore drilling, production, construction and other activities. During the operation of offshore jacket platform at sea, it will be subjected to the periodic action of various loads, which will cause fatigue damage in the internal joints of the structure. To monitor the fatigue damage of structural joints, this paper studies the fatigue analysis method and fatigue crack development rules of typical joints of platforms through fatigue test and finite element analysis. The current study is based on engineering data of J225-IS oil field jacket platform in Bohai Bay, China. Bohai Bay is the largest offshore oil and gas production base in China, among those owned by the China National Offshore Oil Corporation (CNOOC). Based on the engineering data of J225-1S oil field jacket platform in Bohai Bay, three kinds of typical joints are selected in the current research, i.e., T-type plate joint, T-type tube-plate joint and T-type tube joint. Through the fatigue test of joints, the location of fatigue damage and the development of crack damage are observed and analyzed, and the fatigue life test results of joints are recorded. Combining with three kinds of typical joint specimens in fatigue test, this paper establishes the finite element analysis model, calculates, and analyzes the high stress and high strain regions, as well as fatigue damage distribution and fatigue life results of all kinds of joints. By comparing the results of finite element analysis with those of fatigue test, the fatigue damage and fatigue life prediction rules of joints using finite element method are summarized. The results show that the cracks of the three types of joints exist in the weld toe areas, but the fatigue life calculated by the finite element method is less than the experimental results, so the fatigue life prediction by the finite element method is reasonable and safe.

## 2. Fatigue Test

Fatigue strength has a scattering force specific to the four key parameters of the load, design, materials and manufacturing, so it is vital to evaluate the component using experimental tests [95]. JZ25-1S WHPA jacket located in Bohai Bay, China was used in the current study. Figure 1a shows the schematic diagram of JZ25-1S WHPA jacket. The real platform is shown in Figure 1b. The jacket has 6 main legs, 6 piles (the top pile leg of the catheter frame is 16 m × 16 m and is underwater by 6 skirt piles), and 4 secondary legs and piles frame structure are used for supporting the topside weight. The water depths for jacket is 24 meters and the water depth at the side is 19.5 m. The working point is at an elevation (EL) of 7.0 m. The jacket has a tubular steel structure with six main legs and six through the leg piles. The jacket leg consists of two rows of 4 legs spaced at 40 meters to facilitate the floating over installation of the topside. Three horizontal framing levels are used at EL (+) 5.0 m, EL (−) 8.0 m, EL (−) 24.0 m.

### 2.1. Selection of Joints

Fatigue damage of joints caused by harsh environmental conditions on JZ25-1S should be considered. The equipment and accuracy should also be taken into consideration when selecting the measurement method. Based on engineering report of manufacturing company, three types of specimens were used in laboratory measurements: the first one is T-type plate joint prepared by joining two metallic plates of different sizes in such a way that first plate is placed in vertical direction and second smaller plate is placed in horizontal direction and then welded as shown Figure 2a.

The second specimen is the combination of plate and pipe of T-type tube-plate joint prepared by joining a metallic plate and pipe of different sizes in such a way that the longer plate is placed in vertical direction while the smaller pipe is placed horizontally then welded as shown in Figure 2b.

The third specimen used in the testing is the combination of two metallic pipes of different sizes prepared by joining two metallic pipes in T-type joint in such way that the longer pipe is placed horizontally while smaller pipe is placed vertically and then welded as shown in Figure 2c. Furthermore, in Figure 3, the supports (mounts to the base) during the experimental fatigue tests are represented by blue color and the points and directions of the testing force are represented by red color.

Comprehensively considering the platform properties, production conditions, materials, equipment’s loading conditions and capabilities, fatigue tests employ a relatively large load and refers to relevant specifications [96]. Mechanical properties of specimens are given in Table 1.

Carbon dioxide gas shield is used for welding the specimens. Before the parts were connected, the rust, oil stains, oxide scale and other substances harmful to welding on the surface of the steel parts were cleaned with an electric grinder. Appropriate preheating process must be used before steel plate welding to prevent hardened structure and cold cracks. The welding current should not be too large, and the welding speed should not be fast, otherwise it will cause slag removal that will be difficult to remove. The welding parameters are given in Table 2. The welding parameters are provided in ranges. The selection of the magnitudes of the weld parameters from a given range depends on the weld quality requirements while welding each of the specimens. The information about the heat input [kJ/mm] is also provided in Table 2. If there are defects such as pores and slag inclusions after welding, use an electric portable grinder to clean the defective part, and then repair it with manual arc welding. In order to eliminate the welding toe defects, after the specimen is welded, the welding undercut is polished to reduce the influence of the residual stress or the connection, and it is polished with a rotating grindstone. The Q345 steel was selected as the material for test in the current study. The material properties and chemical composition of the Q345 steel are given in Table 1 and Table 3. Low alloy high strength steel Q345 is China GB standard steel and has good properties such as material density of 7.85 g/cm^3^, tensile strength of 470–630 MPa, yield strength of 345 MPa, better than Q235 steel performance. Q345 steel is a low allowable structural steel with fine mechanical properties, low temperature performance, good welding and plasticity, Q345 steel is mainly used in low pressure containers, petroleum tanks, vans, large ships, mechanical components, offshore structures, and high load welded structural parts. The chemical composition of Q345B steel is given in Table 3. All around the world there are a number of other steel grades that are equivalent to the Q345B [97].

### 2.2. Experimental Process 

The complete experimental process is shown in Figure 4. The experimental setup is located in the Marine Engineering Experimental Institute at Zhejiang University. During the start of the experiment, the piezoelectric transducers are first attached to the four different specimens as shown in Figure 5. The specimens are then loaded in the load frame machine, shown in Figure 4. The impedance analyzer was used to check the specimen twice, before applying any load, in order to avoid any possible errors in the results. For detecting cracks in the specimen, a penetration test is applied and the specimen is also analyzed using a digital microscopic camera. The presence of the crack is shown on the computer screen with the help of the VEE Pro software. The experimental procedure is designed based on the guidance provided in the ASTM E466-15 standard force-controlled fatigue tests, ASTM E647-15e1 standard for measurement of fatigue crack growth rates, E739-10(2015) standard for statistical analysis of linear or linearized stress-life (S-N) and strain-life (ε-N) fatigue data and E1049-85(2017) standard for cycle counting in fatigue analysis.

The cyclic load from the load frame is continuously applied on the specimen until the crack is detected. In order to create a more visible surface of specimen for detecting the crack, the DPT-8 dye penetrate inspection material is used to perform the penetration test. The presence of the damage in the specimen will affect the sensitivity of the PZT. PZT is pasted near the welded zone in the case of T-type specimens, in order to achieve better impedance results at the welded zone. The cracks were detected and the damage locations were specified in all the specimens using the microscopic camera. 

### 2.3. Experimental Results

After completing the calibration and preloading of the fatigue testing machine, fatigue test on the specimens were performed. Different specimens are observed and recorded on different regular intervals.

The test results of the three types of joints are relatively different, and the reasons are as follows:(1)Due to the difference in welding quality, the welding strength of each specimen is different. The differences in the quality of the tested joints is mainly dependent on the welding quality.(2)The crack damage location is different. Some are in the weld area, and some are outside the weld area.(3)Due to the existence and influence of the initial defects of the specimen, the experimental results have a certain degree of uncertainty.(4)The fatigue crack damage observed in the experiment may be at different stages.(5)In the current study, the cracks were mainly detected in the heat-affected zone.

Based on the experimental study performed in this paper, the following suggestions are listed for fatigue tests in future research:(1)When making fatigue specimen, the welding conditions and quality of the specimens should be strictly controlled to ensure the consistency of the specimen performance.(2)Different types and positions of structural damage may have an impact on the fatigue results.(3)The impact of initial defects on the structure should be reduced as low as possible.(4)High-precision and sensitive fatigue crack observation devices are suggested to be applied in order to improve the accuracy of observations.

#### 2.3.1. T-Type Plate Joint

During the fatigue loading process of the T-type plate joint, the lower bottom plate was fixed on the base of the test bench by bolts, and the other end was clamped on the fixture that apply periodic cyclic load. The loading frequency was 1 Hz and magnitude was 150 kN. The testing machine was loaded from –150 kN to +150 kN. The load value used in the experiment is much larger than the actual operating conditions, with the aim of producing fatigue cracks within a limited experimental time.

The damage and cracks were observed and monitored under certain period. Once the fatigue crack damage was detected, the loading was stopped, and the number of loading cycles at that time was recorded. Crack location and damage situation of T-type plate joint were observed, the results are summarized in Table 4.

Due to the fluctuation of the fatigue life observed in the tests, the average fatigue life is selected in this paper. The available fatigue life of welded metallic plate joint is 16,402 times. As shown in Figure 6, the fatigue cracks of T-type plate joint often occur in the weld toe area on the joint connection part. The damage location of the weld toe is more concentrated in the location near the edge of the side corner. It can be seen that fatigue crack damage is prone to be shown in the upper welding toe of the joint connection weld, especially the area near the edge corner, which should also be the focus in the engineering practice.

While observing the fatigue crack damage of the specimens, the early fatigue cracks of the typical joints can be found. Taking G10 as an example, when sub-millimeter crack is shown in the microscopic observation, open the test machine dynamic loading. The crack develops from 0.463 mm, gradually expanding to 0.687 mm, and then growing to over 1.846 mm. 

In the process of gradual expansion and extension, it can be seen that the cracks change from a simple straight line at the initial stage of small microcracks to a bent and broken line shape. On the other side, some cracks appeared in other areas were observed. As the fatigue loading continued to apply, obvious fatigue fractures appeared at the welds and the edges of the base metal, as shown in Figure 7.

#### 2.3.2. T-Type Tube-Plate Joint

The loading frequency of T-type tube-plate joint test was 1 Hz and magnitude was 150 kN. The testing machine is loaded from –150 kN to +150 kN. The load value used in the experiment is much larger than the actual operating conditions, with the aim of producing fatigue cracks within a limited experimental time.

The damage and cracks were observed and monitored under certain period, and the results were summarized in Table 5. As shown in Table 5, in the current study, cracks were also observed in the welds.

In order to reduce the error, the average fatigue life is selected as the test results in this paper. The available fatigue life of T-type tube-plate joint is 11,856 times. The fatigue cracks of T-type tube-plate joint often occurred in the weld area of the tube-plate connection. Most cracks appeared in the upper weld toe part of the connecting zone, as shown in Figure 8a,b, and a small number of fatigue cracks appeared in the lower weld toe, as shown in Figure 8c,d. Therefore, attention should be paid to the weld toe of the connection zone, especially the upper weld toe area.

H10 specimen was continuously loaded after recording the corresponding fatigue cycles when the fatigue crack appears. As shown in Figure 9, a small 0.559 mm sub-millimeter crack, like a straight line is first observed. Then the crack is expanded to 0.929 mm after 1000 cycles, growing later on to nearly 2.424 mm. In the process of crack propagation, the shape of the crack mainly changes linearly. With an increase in the number of loading, the fatigue crack has a relatively obvious growth, and gradually expands to a larger visible scale.

#### 2.3.3. T-Type Tube Joint

The loading frequency of T-type tube joint test was 1 Hz and magnitude was 180 kN. The testing machine was loaded from –180 kN to +180 kN. As compared to the other two types of joints, the load magnitude is more due to the size of the weld section. The damage and cracks were observed and monitored under certain period, and the results are summarized in Table 6.

In order to reduce the error, the average fatigue life is selected as the test results in this paper. The available fatigue life of T-type tube joint is 449 times. As shown in Figure 10, the fatigue cracks of T-type tube joint often occurred in the weld area of the tube-tube connection. Crack damage often occurred in the upper and lower toe areas and extended along the welding direction. T-type tube joints are relatively common in jacket platforms. As one of the main structural components, it is necessary to focus on the weld area, especially the upper and lower welding toe, should be carefully inspected and monitored in the engineering practice.

Figure 11a shows the first fatigue crack observed (0.864 mm) during the test, which is a linear crack. As the fatigue loading continues, the crack expands in a linear shape, and new cracks are also generated, as shown in Figure 11b.

## 3. Finite Element Analysis

### 3.1. Modelling

Referring to the typical joints that have completed the fatigue test in Section 2, finite element models were established, and stress analysis was carried out using ANSYS to study the fatigue life prediction. The three-dimensional modeling software SolidWorks was used to establish the three-dimensional model of specimens and welded seam. Since the welds of specimens were all polished after welding, the weld geometry is consistent with the actual weld seam, as shown in Figure 12. Shared nodes were used to connect the welded seam and base metal.

The materials of three the typical joints are all Q345 steel. The performance parameters of Q345 are shown in Table 7.

### 3.2. Mesh

The geometric models of the three typical joints have symmetrical characteristics. Therefore, the 1/4 model can be used in the finite element analysis, which can reduce computational costs and improve the efficiency and accuracy of finite element calculation.

The hexahedral mesh with Solid 186 elements was used, which are second-order elements. Compared with linear elements, it has more integration points so that the results are more accurate. The welds are locally refined, and the mesh model information of each joint is shown in Table 8.

### 3.3. Boundary Conditions and Applied Load

In the analysis, the constrained boundary of the finite element model is to simulated installation of specimens on the fatigue bench. Detailed boundaries and loads are described as follow:(1)T-type plate joint: the boundary conditions are set as shown in Figure 13a. The A end is fixed and the B end is loaded.(2)T-type tube-plate joint: the boundary conditions are set as shown in Figure 13b. The A end is fixed, and the B end is loaded.(3)T-type tube joint: The boundary conditions are set as shown in Figure 13c. Both sides of the lower pipe are fixed to simulate clamping, and the B end is loaded.

The load application is also consistent with the fatigue test. When the finite element model is a 1/4 symmetric model, the load is also 1/4 of the total load. The load application of each joint is shown in Table 9.

### 3.4. S-N Curve

Fatigue strategy of offshore structures is founded on the S-N curves obtained from tubular joint testing, where faults are defined as wall thickness penetration [96,97]. However, in practice, fatigue cracks are probable to continue to raise around the welding environment after cracking through the walls, and in some cases, significant residual life has been found after cracking through thickness [98,99,100,101,102]. Most fatigue calculations are carried out from the S-N curves. These curves have been shown to be a proper decent approximation of fatigue life. S-N is a short form for stress and period/cycle (N), respectively [103]. The S-N curve is usually best suited for high cycle fatigue, where the strain remains within the elastic range. The low-cycle fatigue often causes stress in the plastic range and is best described using a strain-based approach [103].

The S-N curve uses the fatigue strength of the standard specimen as the ordinate, and the logarithm of the fatigue life as the abscissa, which represents the relationship between the fatigue strength and fatigue life of the standard specimen under certain cycle characteristics. The S-N curves need to be selected before finite element simulation. The fatigue S-N curve formula is defined as follows:(1)$$\log N=\log\bar{a}-\mathrm{mlog}(\Delta \sigma {\left(\frac{t}{t_{\mathrm{ref}}}\right)}^{k})$$
(2)$$\log\bar{a}=\log a-2S_{\mathrm{logN}}$$
where N is the fatigue life under stress amplitude Δσ (unit: times), m is the slope of the S-N curve, $\log\bar{a}$ is the intercept of the curve on the logarithmic N axis, loga is the intercept of average S-N curve, $S_{\mathrm{logN}}$ is the standard deviation of the logarithm N, which is 0.2. ${\left(\frac{t}{t_{\mathrm{ref}}}\right)}^{k}$ is the correction coefficient considering the wall thickness. Under the wall thickness conditions in this paper, the coefficient is taken as 1.0. Table 10 lists the S-N curve category for the three T-type tube-plate joints. Figure 14 shows S-N curves for the three typical joints given in Table 10. As shown in Figure 14, the fatigue strength is maximum for the T-type plate joint, given in Figure 14a, followed by the T-type tube joint, as shown in Figure 14c and the fatigue strength is minimum for the T-type tube-plate joint as shown in Figure 14b.

Based on the model established in Section 3.1 stress analysis of each typical node and fatigue analysis of three nodes are carried out. Finally, the fatigue test results of the test pieces are compared and analyzed.

### 3.5. Discussion

#### 3.5.1. Fatigue Damage Location

(1)T-type plate joint

It can be seen from Figure 15 that the upper weld toe area is the location of the maximum fatigue damage and has shortest fatigue life. Out of the 14 sets of fatigue tests, 9 specimens have fatigue crack damage in the upper weld toe and its vicinity, and other specimens were cracked in the lower weld toe area.

(2)T-type tube-plate joint

It can be seen from Figure 16 that the upper weld toe is the location of the largest fatigue damage and has the shortest fatigue life, and part of the lower weld toe is also the location of the largest fatigue damage. In the fatigue test, six specimens were found to have fatigue crack damage in the upper weld toe and its vicinity, and two specimens were found to have crack damage in the lower weld toe area.

(3)T-type tube joint

It can be seen from Figure 17 that the lower weld toe is the location of the largest fatigue damage and has the shortest fatigue life. Part of the upper weld toe is also the location of the largest fatigue damage. In the fatigue test, fatigue cracks were found in the lower weld toe and its vicinity in four specimens, and cracks were found in the upper weld toe area in three specimens.

#### 3.5.2. Fatigue Life

The fatigue test data of each type of specimen is statistically analyzed to obtain the test fatigue life, which is compared with the life obtained from the finite element analysis. The results are shown in Table 11. The finite element analysis method considers the fatigue cumulative damage analysis and the SN curve in the DNV specification; therefore, the results of the finite element analysis have large safety margins. The finite element results in Table 11 are less in magnitude than the life of the fatigue tests, which also proves that the finite element results are conservative. The large difference in the fatigue life, in Table 11, is due to the various assumptions in the modeling process. Due to its under estimation of the fatigue life, the difference is even helpful to achieve maximum safety, if the finite element modeling is used for the calculation of the fatigue life. Among them, the finite element results of T-type tube joint are small and close to those of the test, indicating that the finite element method is safe and accurate for the life prediction analysis of such joints. The results of the finite element calculation of T-type plate joint and T-type tube-plate joint are much smaller than the test results. The difference in the magnitude of the results may also be due to the hysteresis of fatigue cracks observed in the test. On the other hand, it shows that the results of finite element analysis of these three types of joints are conservative and have a relatively large safety margin.

In summary, the results related to the damage location of the finite element analysis are in good agreement with the experimental test results. The fatigue life calculated by the finite element method is relatively small compared to the experimental test results. In the future, it is feasible and safe to predict the fatigue life by the finite element method. 

As compared to the other studies in the literature, the current study is focused on an actual oil field jacket platform in Bohai Bay, China. Many of the studies in the literature are focused either on the experimental or on the numerical modeling but in the current study both the experimental and numerical modeling is performed and in the end, the results are compared. The comparison acts as a validity of the analysis performed in the current study. While most of the studies in the literature are focused on one or two types of joints, the current study is focused on the main three kinds of joints that are found in the jacket platforms, i.e., T-type plate joint, T-type tube-plate joint, and T-type tube joint. Through the fatigue test of joints, the location of fatigue damage and the development of crack damage are observed and analyzed in the current study and the fatigue life test results of joints are recorded. With the three kinds of typical joint specimens in fatigue test, the current study establishes the finite element analysis model, calculates, and analyzes the high stress and high strain regions, as well as fatigue life and fatigue damage distribution.

## 4. Conclusions

Based on the engineering data of J225-1S oil field jacket platform in Bohai Bay, three kinds of typical joints, i.e., T-type plate joint, T-type tube-plate joint, and T-type tube joint are selected for investigation in the current study. Through the fatigue test of joints, the location of fatigue damage and the development of cracks were observed and analyzed, and the fatigue life test results of joints were recorded. The outcomes of the current study are summarized below:(1)In the fatigue test, the fatigue crack damage can be located relatively accurately through the stained flaw detection method, combined with the digital microscope to observe the crack damage of the specimen, which can detect micro scale cracks more effectively. A crack of the size 0.463 mm is detected in the current study.(2)In the early stage of fatigue crack development of joints, fatigue damage is often linear small-scale cracks (crack lengths between 0.463 mm and 14.640 mm are detected in the current study), and the propagation direction often expands along the welding direction. Early cracks when grows more are changed in a broken line shape, and slowly change after reaching a certain size, then other new cracks are also formed.(3)The fatigue cracks of the three types of joints are the first to appear at the weld toe and its vicinity, and they spread along the welding direction. The fatigue damage location of T-type plate joint and T-type tube-plate joint is more concentrated in the upper weld toe area, and the fatigue damage location of the T-type tube joint is closer to the lower weld toe area. In engineering practice, the above-mentioned fatigue damage areas should be taken into consideration.(4)Through finite element analysis, the high stress and strain areas of the three types of typical joints are determined. The dangerous areas appear in the weld area of the joint connection. After comparing the sensitive areas of the above joints. The test results are basically consistent, which proves the reliability of the finite element method in analyzing the fatigue damage location.(5)The finite element calculation results and the test results are compared and analyzed. The results of the T-type tube joint are closer, and the results of T-type plate joint and T-type tube-plate joint are relatively small and still have a relatively large safety margin, which proves that the use of finite element analysis to predict fatigue life will provide large safety margin and thus it is more safe to predict the fatigue life of the joints by finite element method in engineering applications.

## Figures and Tables

**Figure 1 sensors-21-03294-f001:**
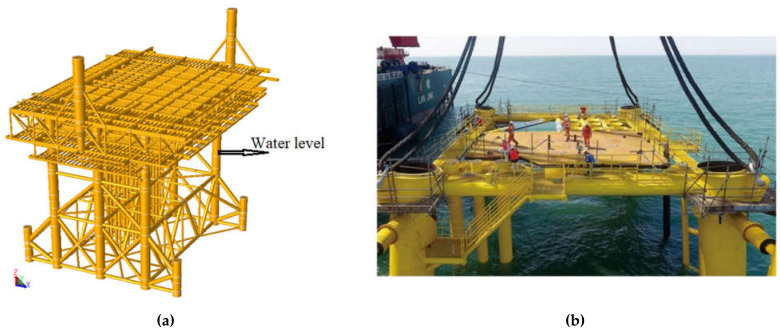
(**a**) Schematic diagram of JZ25-1S jacket; (**b**) real platform.

**Figure 2 sensors-21-03294-f002:**
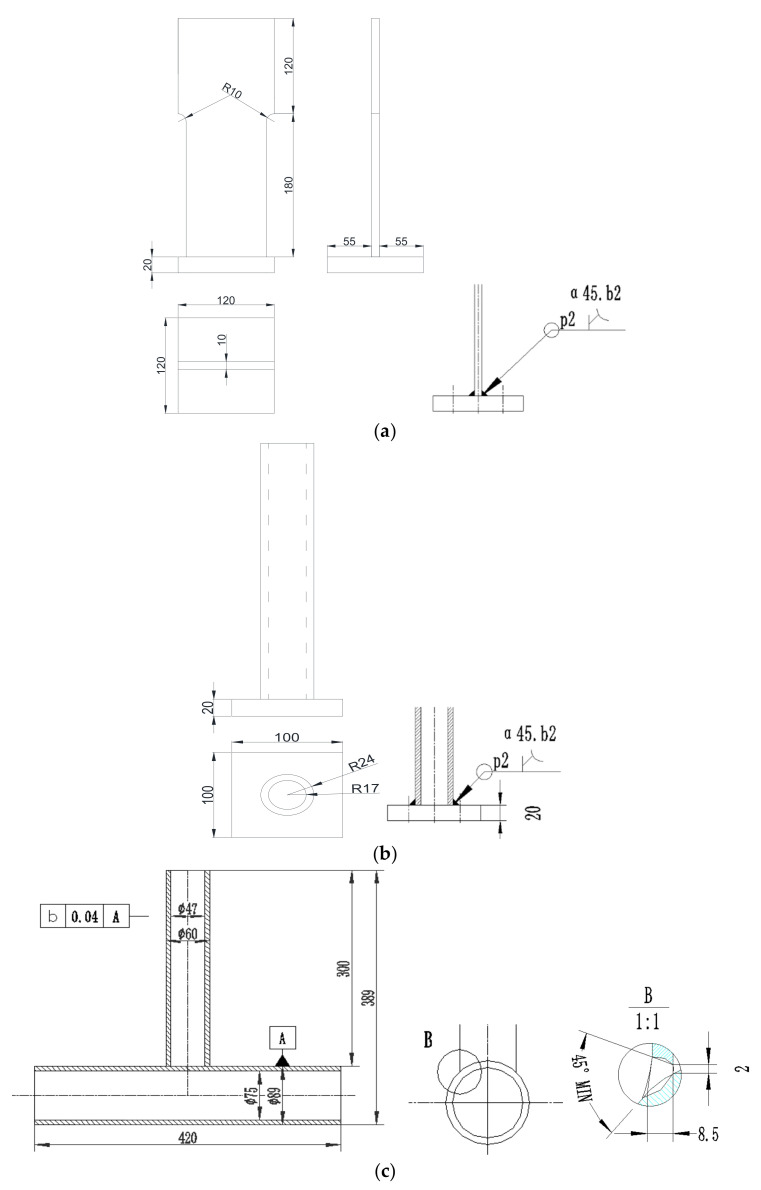
Geometry of the three types of specimens used in laboratory measurements. (**a**) T-type plate joint. (**b**) T-type tube-plate joint. (**c**) T-type tube joint.

**Figure 3 sensors-21-03294-f003:**
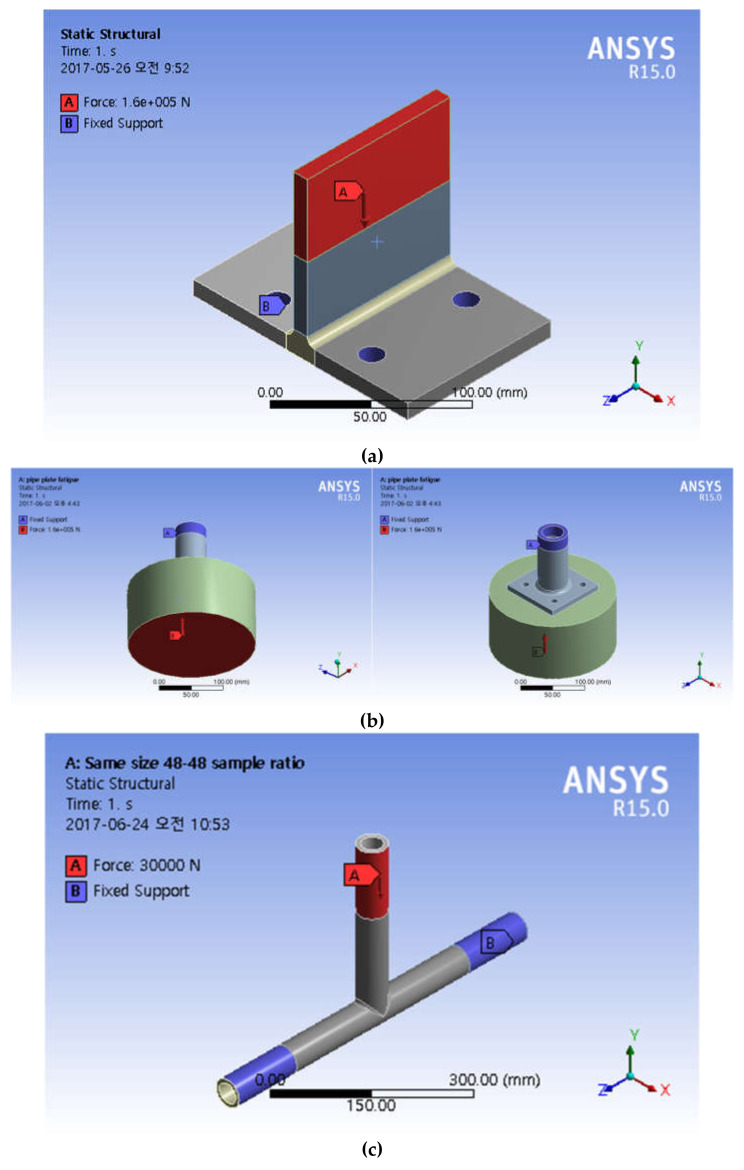
Supports and the points and directions of the testing force (**a**) for static structures (**b**) for pipe plate fatigue (**c**) for same size sample ratio.

**Figure 4 sensors-21-03294-f004:**
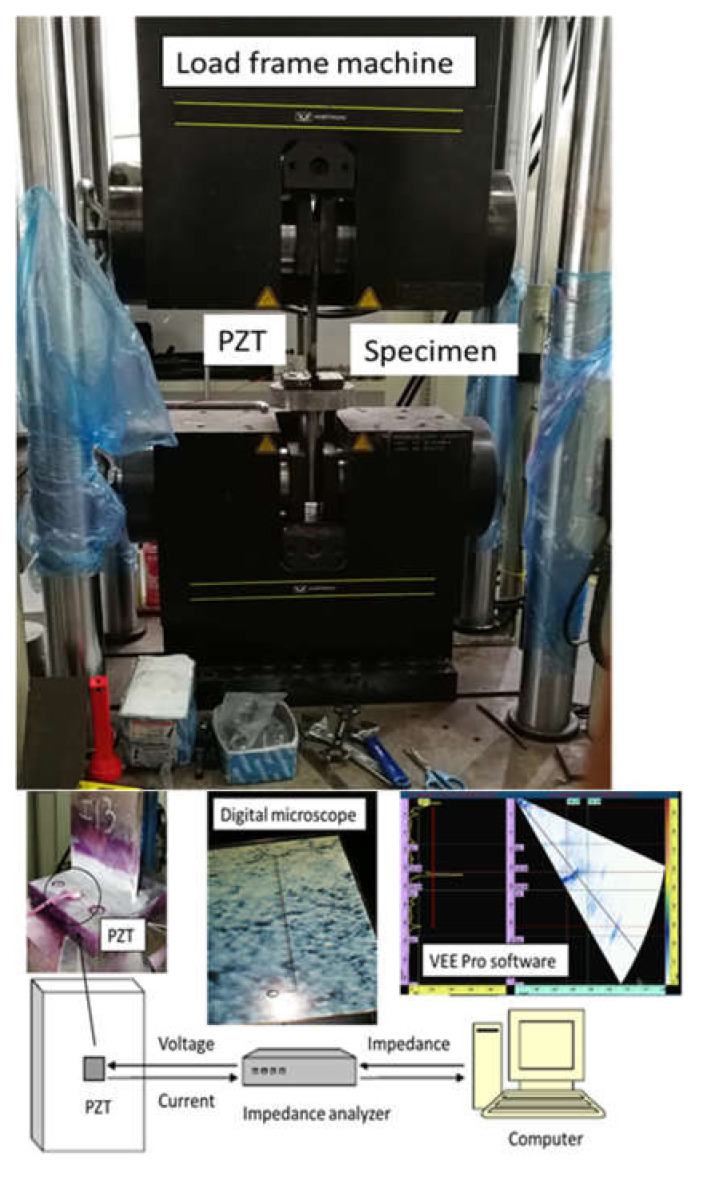
Experimental setup.

**Figure 5 sensors-21-03294-f005:**
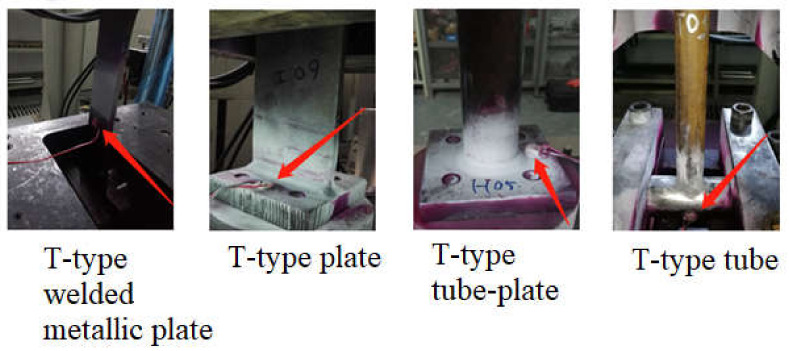
PZT transducers mounted on the specimens.

**Figure 6 sensors-21-03294-f006:**
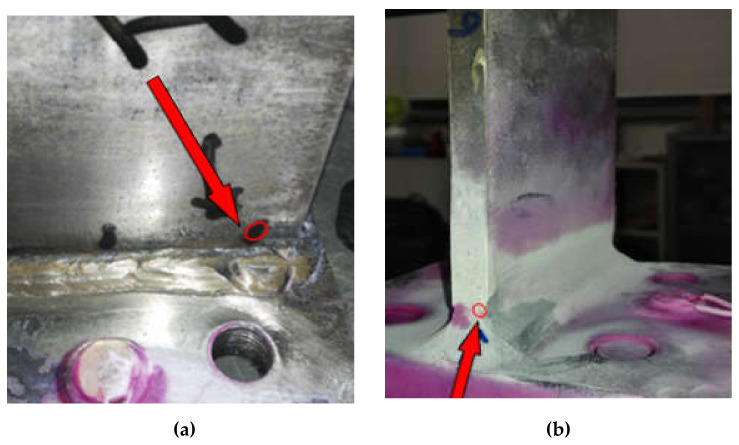
Cracks locations on T-type welded metallic plate (**a**) at upper welding toe (**b**) near the edge corner (**c**) on the weld toe (**d**) near side edge corner.

**Figure 7 sensors-21-03294-f007:**
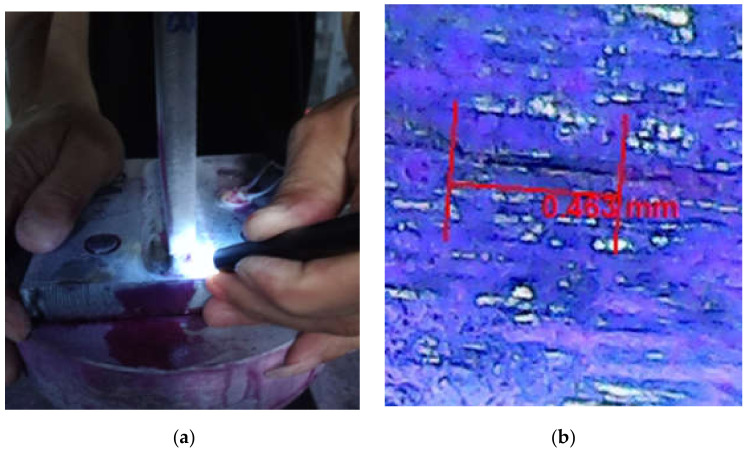
Microscopic observation of fatigue crack in T-type plate joint (**a**) cracks initiated at edges of the base metal (**b**) observed crack of 0.463 mm length (**c**) observed crack of 0.687 mm length (**d**) observed cracks of 0.561 mm and 1.285 mm lengths.

**Figure 8 sensors-21-03294-f008:**
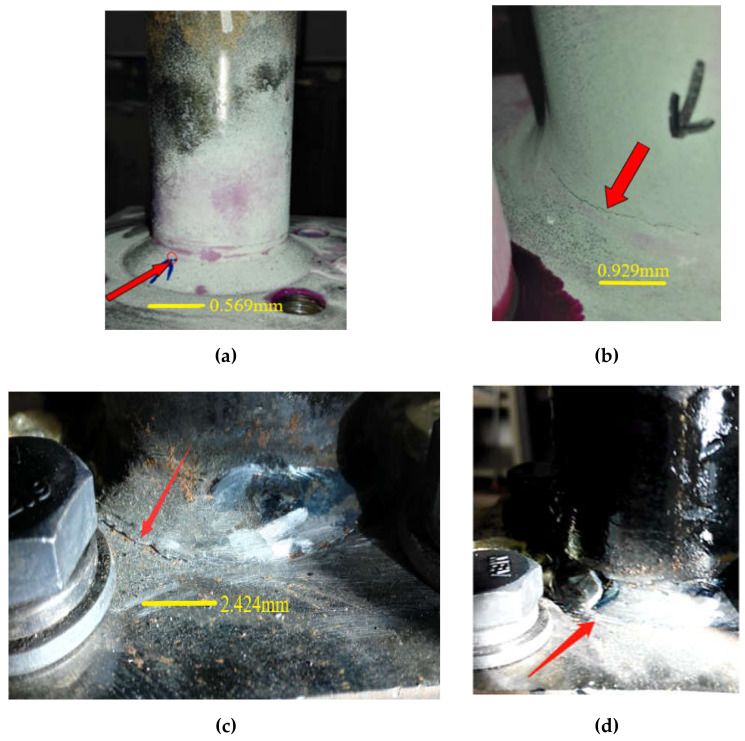
Cracks of T-type tube-plate joint (**a**) observed crack of 0.569 mm length (**b**) observed crack of 0.929 mm length (**c**) observed crack of 2.424 mm length (**d**) large crack initiated at edges of the base metal.

**Figure 9 sensors-21-03294-f009:**
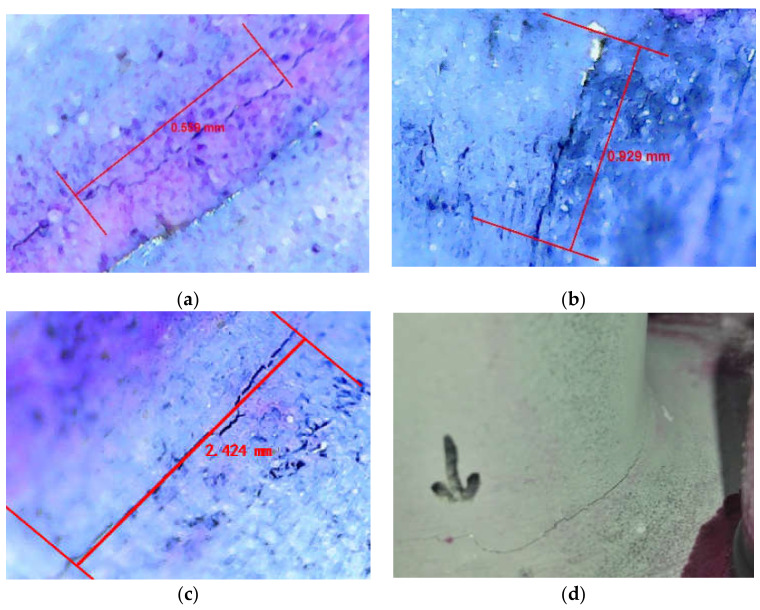
Microscopic observation of fatigue crack in T-type tube-plate joint (**a**) observed crack of 0.559 mm length (**b**) observed crack of 0.929 mm length (**c**) observed crack of 2.424 mm length (**d**) cracks initiated at edges of the base metal.

**Figure 10 sensors-21-03294-f010:**
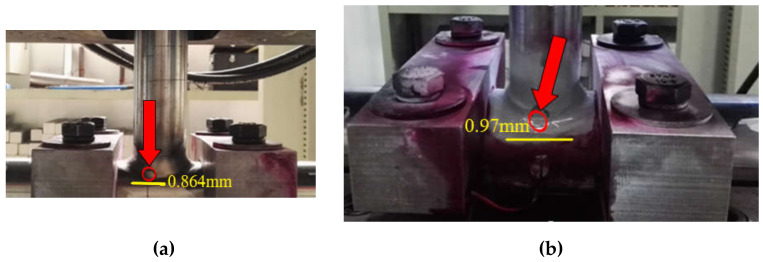
Cracks of T-type tube joint (**a**) observed crack of 0.864 mm length (**b**) observed crack of 0.970 mm length.

**Figure 11 sensors-21-03294-f011:**
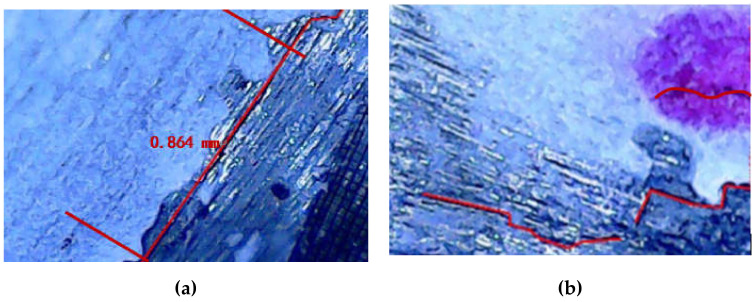
Microscopic observation of fatigue crack in T-type tube joint (**a**) observed crack of 0.864 mm length (**b**) cracks initiated at edges of the base metal.

**Figure 12 sensors-21-03294-f012:**
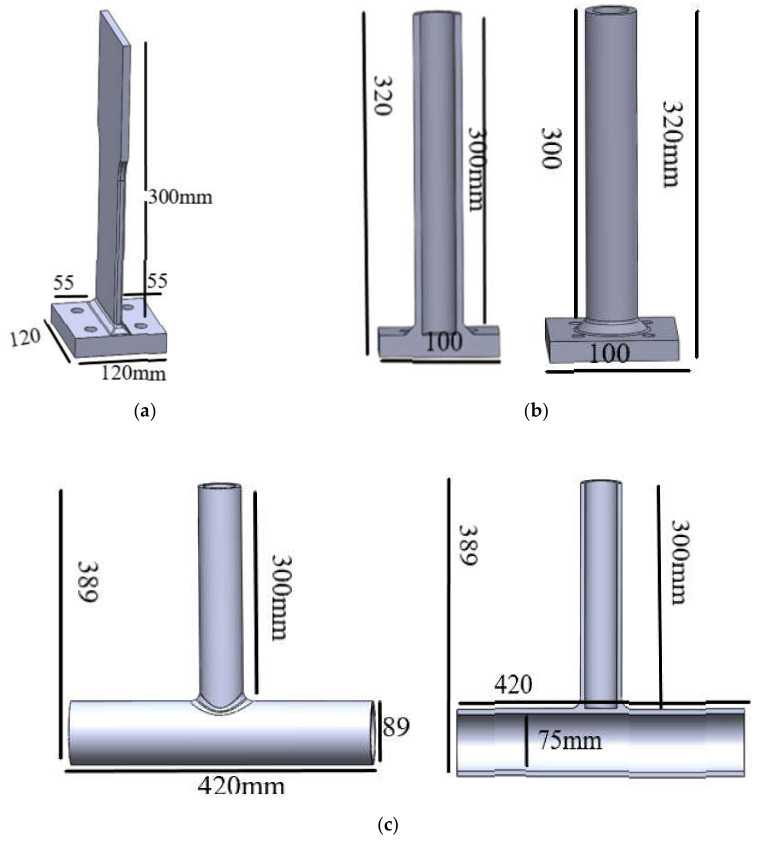
Finite element models of three typical joints (**a**) T-type plate joint (**b**) T-type tube-plate joint (**c**) T-type tube joint.

**Figure 13 sensors-21-03294-f013:**
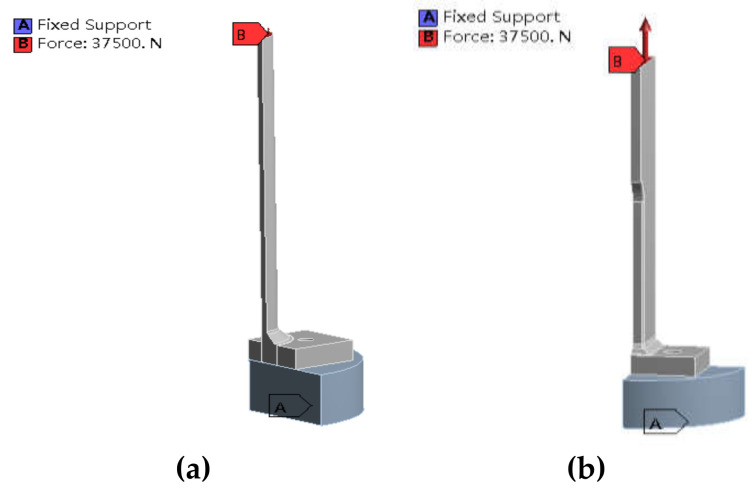
Boundary condition of each joint (**a**) T-type plate joint (**b**) T-type tube-plate joint (**c**) T-type tube joint.

**Figure 14 sensors-21-03294-f014:**
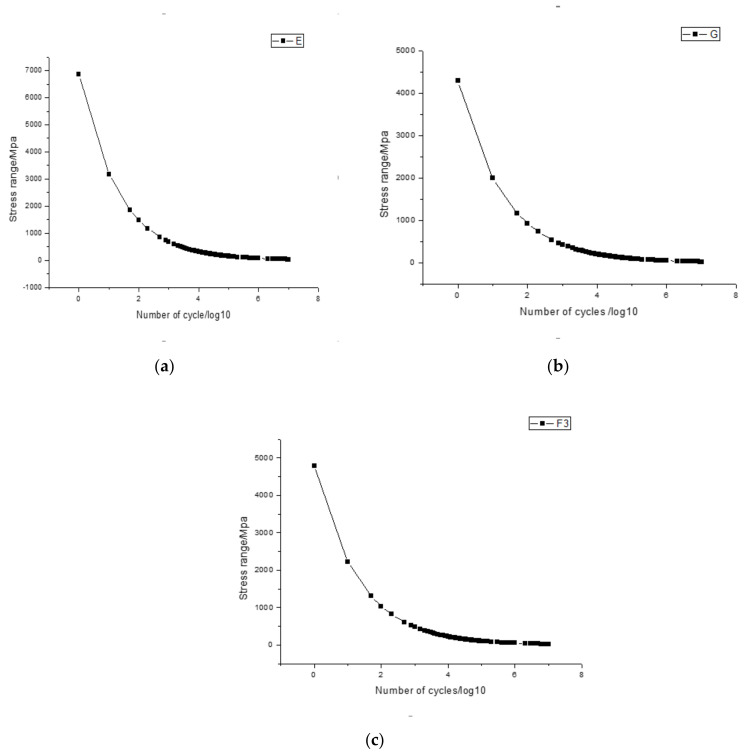
S-N curves for three typical joints. (**a**) Curve E for T-type plate joint. (**b**) Curve G for T-type tube-plate joint. (**c**) Curve F3 for T-type tube joint.

**Figure 15 sensors-21-03294-f015:**
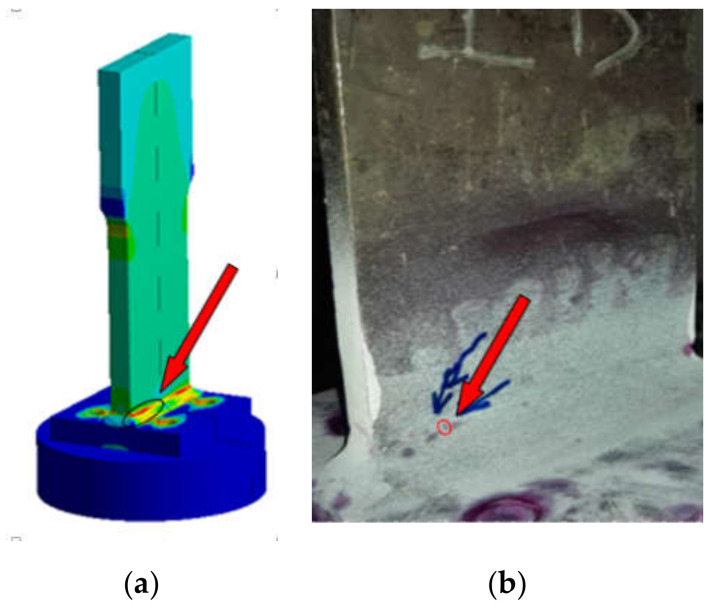
Comparison of fatigue damage in T-type plate joint (**a**) Finite element result (**b**) Test result.

**Figure 16 sensors-21-03294-f016:**
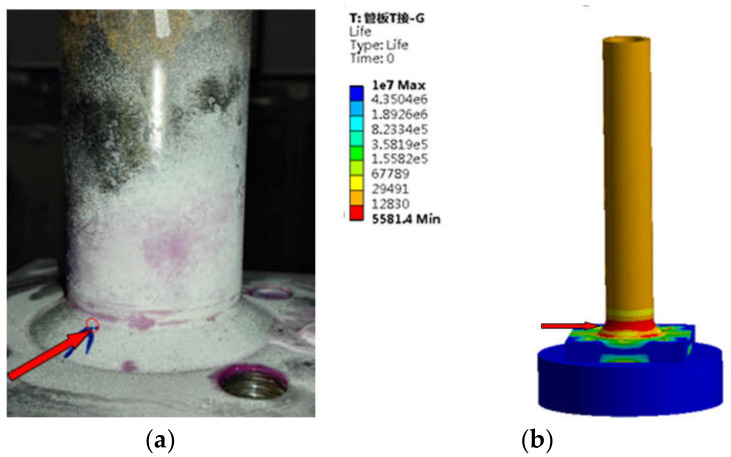
Comparison of fatigue damage in T-type tube-plate joint (**a**) Test result (**b**) Finite element result.

**Figure 17 sensors-21-03294-f017:**
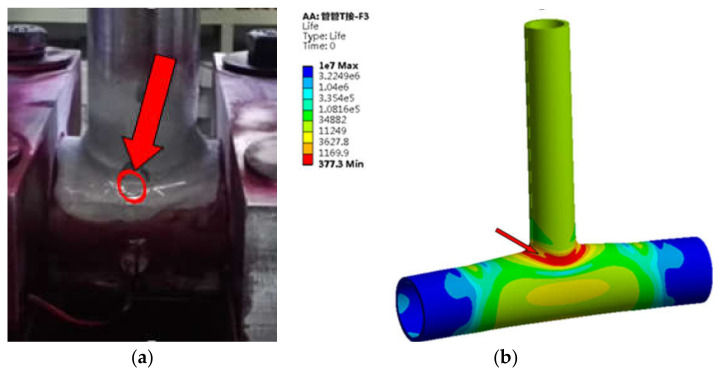
Comparison of fatigue damage in T-type tube joint (**a**) Test result (**b**) Finite element result.

**Table 1 sensors-21-03294-t001:** Mechanical properties of specimens.

Property	Value
Material types Tensile strength (MPa)	Steel (Q345B) 490–675
Yield strength (MPa)	≥345
Elongation after fracture	≥21%

**Table 2 sensors-21-03294-t002:** Welding parameters.

Specimen No	Welding Current [A] 50–100	Welding Voltage [V] 18–21	Welding Speed [mm/s] 5–20	Heat Input [kJ/mm]
1	90	20.0	6.5	0.28
	75	19.0	5.0	0.29
2	85	19.5	5.35	0.31
	94	20.5	6.5	0.30
3	98	18.5	5.0	0.37
	88	21.0	6.18	0.30

**Table 3 sensors-21-03294-t003:** Chemical composition of Q345B steel.

Elements	Content
C≤ Mn	0.2 1.0–1.6
Si≤	0.035
Al≥	0.02–0.15
Nb	0.015–0.06

**Table 4 sensors-21-03294-t004:** Test results of T-type plate joint.

Specimens	Load (kN)	Fatigue Life (Times)	Damage Position	Crack Length (mm)
G01	150	9409	Weld toe	13.500
G02	150	6666	Weld toe	9.760
G03	150	5361	Weld seam	Large crack
G04	150	976	Weld seam	Large crack
G05	150	4714	Weld toe	1.360
G06	150	10,017	Weld toe	2.816
G07	150	6937	Weld toe	5.390
G08	150	2894	Bolt	Large crack
G09	150	3450	Weld toe	2.254
G10	150	17,114	Weld toe	0.463
G11	150	24,202	Weld toe	8.386
G12	150	43,279	Weld toe	6.850
G13	150	62,441	Weld toe	Large crack
G14	150	32,177	Weld toe	1.038

**Table 5 sensors-21-03294-t005:** Test results of T-type tube-plate joint.

Specimens	Load (kN)	Fatigue Life (Times)	Damage Position	Crack Length (mm)
H01	150	3553	Near weld seam	9.940
H02	150	1400	Bolt cracked	12.680
H03	150	2350	Weld seam cracked	Snapped
H04	150	5491	Weld seam	13.000
H05	150	12,446	Weld toe	4.200
H06	150	8393	Weld toe	1.050
H07	150	10,758	Weld toe	3.468
H08	150	2465	Weld toe	0.920
H09	150	17,167	Weld toe	14.640
H10	150	5429	Weld toe	0.559
H11	150	15,491	Weld seam cracked	Snapped
H12	150	28,841	Weld toe	2.480
H13	150	21,810	Weld toe	10.129
H14	150	30,391	Snaped	Large crack

**Table 6 sensors-21-03294-t006:** Test results of T-type tube joint.

Specimens	Load (kN)	Fatigue Life (Times)	Damage Position	Crack Length (mm)
T01	180	638	Outside weld toe	12.470
T02	180	337	Weld toe	8.984
T03	180	794	Weld toe	4.247
T04	180	618	Weld toe	0.864
T05	180	529	Weld toe	3.766
T06	180	224	Weld toe	2.443
T07	180	321	Weld toe	4.596
T08	180	132	Weld toe	0.947

**Table 7 sensors-21-03294-t007:** Performance parameters of Q345.

Material	Young’s Modulus (GPa)	Poisson Ratio	Tensile Strength (MPa)	Yield Point (MPa)
Q345	206	0.28	550	320

**Table 8 sensors-21-03294-t008:** Mesh information of each joint.

Type	Grid Size (mm)	Number of Grid Elements	Number of Grid Nodes
T-type plate joint	2	60148	266232
T-type tube-plate joint	2	69457	307746
T-type tube joint	2	43546	211881

**Table 9 sensors-21-03294-t009:** Loads for each joint type.

Type	Finite Element Model	Test Load (N)	Simulative Load (N)
T-type plate joint	1/4 model	150,000	37,500
T-type tube-plate joint	1/4 model	150,000	37,500
T-type tube joint	1/4 model	180,000	47,500

**Table 10 sensors-21-03294-t010:** Three T-type tube-plate joints correspond to the S-N curve category.

The Type of Connection Node	S-N Curve Category	The Thickness Factor Is Corrected
T-type plate joint	E	Without
T-type tube-plate joint	G	Without
T-type tube joint	F3	Without

**Table 11 sensors-21-03294-t011:** Comparison between the outputs of the fatigue tests and finite element analysis method.

Type	Test Results (Times)	FE Results (Times)	Fatigue Standard Deviation	Ratio of Finite Element Life to Test Life
T-type plate joint	16402	9113	18157.62	55.56%
T-type tube-plate joint	11856	5581	8400.58	47.07%
T-type tube joint	449	377	295.11	83.96%

## Data Availability

The data are available by contacting the corresponding author.

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
