# Peer review of "Fatigue Crack Monitoring of T-Type Joints in Steel Offshore Oil and Gas Jacket Platform"

_sensors, 2021, doi:10.3390/s21093294_

Round 1

Reviewer 1 Report

This is a resubmission of a previously reviewed and rejected manuscript.  After reading the revised version and the responses of the authors, I uphold my generally positive view of the manuscript, but I have also a feeling that most of the reviewer remarks have been treated superficially. In the case of my review, the authors decided to address and implement remarks no 1-5, but remarks no 6-12 are largely ignored and shunted off. My recommendation is still a minor revision.

Remarks:

1) The authors have added Table 9 and Fig. 12. They specify S-N curves up to 10^8 parameters. Is it really necessary? The largest number of cycles in the present manuscript is <10^5 and in the case of one joint <10^3. Moreover, Table 9 seems to be copied and pasted from another source, and it even refers to a non-existing equation.

2) Please specify directly what type of a PZT transducer is used. Please show on the scheme, where it was mounted to the specimens.

3) Please clearly mark on specimen schemes (e.g., Fig.2):
a) supports (mounts to the base) during the experimental fatigue tests
b) points and directions of the testing force
Fig.11 is much further into the manuscript, and throughout most of the lecture the reader has no clue about these important details.

4) Table 3: does it make sense to specify the lengths of the cracks up to 1 um accuracy? [Yes, the unit is mm, but the accuracy is 1 um.]

5) The average fatigue lifes (in loading cycles) are stated for the three types of specimens as average values. However, the individual values are widely scattered. It would be helpful to state the standard deviation besides the average value.

6) Table 9: please list also the standard deviation of the experimetnal results.

7) Why the load amplitude for the tube joint (sect.2.3.3) was larger than for the other two types of joints? It was 180 kN vs. 150 kN. It is astonishing, because the much shorter fatigue life (Table 5) might suggest a smaller loading amplitude instead of a larger one.

Author Response

The following represent point-by-point answers to the reviewers’ comments. Appropriate revisions are made in the revised manuscript, as explained hereunder. In the revised version of the manuscript, all the revisions are highlighted in yellow.

Reviewer # 01 Comments:

This is a resubmission of a previously reviewed and rejected manuscript. After reading the revised version and the responses of the authors, I uphold my generally positive view of the manuscript, but I have also a feeling that most of the reviewer remarks have been treated superficially. In the case of my review, the authors decided to address and implement remarks no 1-5, but remarks no 6-12 are largely ignored and shunted off. My recommendation is still a minor revision.

Response: The authors are thankful to the reviewer for the valuable comments. As suggested, in the revised version of the manuscript, all the comments are properly addressed. The following represent, point-by-point answers to the reviewers’ comments.

  • The authors have added Table 9 and Fig. 12. They specify S-N curves up to 10^8 parameters. Is it really necessary? The largest number of cycles in the present manuscript is <10^5 and in the case of one joint <10^3. Moreover, Table 9 seems to be copied and pasted from another source, and it even refers to a non-existing equation.

Answer: The authors are thankful to the reviewer for the valuable comment. As suggested, in the revised version of the manuscript, the Table 9 and Fig. 12 have been removed.

  • Please specify directly what type of a PZT transducer is used. Please show on the scheme, where it was mounted to the specimens.

Answer: In the current study the PTZ-51 ballast ceramic sheet, s15mm, 1MHz frequency, one-sided lead type PZT transducer is used. As suggested, the locations where the PZT transducers are mounted on specimens is shown in Figure 5, on page 10 of the revised manuscript.

  • Please clearly mark on specimen schemes (e.g., Fig.2):
  1. a) supports (mounts to the base) during the experimental fatigue tests
  2. b) points and directions of the testing force

Fig.13 is much further into the manuscript, and throughout most of the lecture, the reader has no clue about these important details.

Answer:  As suggested, in the revised version of the manuscript, the supports (mounts to the base) during the experimental fatigue tests and the points and directions of the testing force are shown in Figure 3, on page 7 of the revised manuscript. In the revised version of the manuscript, an explanation is added for Fig.13, on page 19 of the manuscript. Fig.13 is given as Fig.14 in the revised manuscript.

  • Table 3: does it make sense to specify the lengths of the cracks up to 1 um accuracy? [Yes, the unit is mm, but the accuracy is 1 um.]

Answer: The authors are thankful to the reviewer for the valuable comment and for this excellent observation. Yes, the unit is mm, but the accuracy is 1 um, and we used an electron microscope to measure it. Table 3 is now given as Table 4 in the revised manuscript.

  • The average fatigue lifes (in loading cycles) are stated for the three types of specimens as average values. However, the individual values are widely scattered. It would be helpful to state the standard deviation besides the average value.

Answer: The authors are thankful to the reviewer for the valuable comment. As suggested, in the revised version of the manuscript, the standard deviation for the three types of specimens are listed in Table 11.

  • Table 9: please list also the standard deviation of the experimental results.

Answer: The authors are thankful to the reviewer for the valuable comment. As suggested, in the revised version of the manuscript, the standard deviation of the experimental results are listed in Table 11.

  • Why the load amplitude for the tube joint (Sect.2.3.3) was larger than for the other two types of joints? It was 180 kN vs. 150 kN. It is astonishing, because the much shorter fatigue life (Table 5) might suggest a smaller loading amplitude instead of a larger one.

Answer: As compared to the other two types of joints, the load magnitude is more for the tube joint, due to the size of the weld section. This explanation is added on page 15 of the revised manuscript.

Finally, the authors wish to thank the reviewer for his constructive remarks, which are well-taken and implemented to improve the clarity and quality of the manuscript. We thank you for the time you put in reviewing our paper and look forward to meet your expectations.

Reviewer 2 Report

Dear Authors,

I have reviewed paper "Fatigue Crack Monitoring of T-type Joints in Steel Offshore Oil and Gas Jacket Platform".

Presented investigations fulfill the aims and scope of the journal. My suggestions are listed below.

General remarks:

  • You have presented 41 references. Only three of them have been published after 2018. In my opinion you should present more of the newely published articles. The science made big step forward last years, and it should be visible in your paper.

Introduction:

  • As you stated, the offshore structures may undergo faiures. However, you have not mentioned any about repairs. There are two ways, one, took the stucture outsside the water, and second, performin welding repairs in the water. Are there differences? I think it is wortk to mention about techniques allowing to perform repairs in the water (e.g., additional stitches, temper bead welding, changing the bead sequence, waterproof coating on the surface of filler materials).
  •  You mentioned about cracks, which are present in offshore structures. You have focused on the fatigue cracks. However, in offshore structures made from HSLA steels the most important problem is cold cracking. Please add this information.

Fatigue Test:

  • Please add information about dimensions of investigated JZ25-1S WHPA jacket.
  • Please name the welding method used in your investigations.
  • It is worth to show the chemical composition of Q345B steel.
  • Table 2 - presented range of parameters are very wide. Please commet this table. When you have used low parameters, when higher. Also, the information about heat input [kJ/mm] shoud be presented. This is more important factor than others. Please calculate and show these values.
  • 2.2. - have you used any rules from relevant standards? If yes, please mention the number of these standards. If not, please add information why.
  • 2.3. why there were differences in the quality of tested joints? Please comment.
  • Line 183 - why these values were used? Are they corresponding with real conditions? It should be straightly marked in your text. The same comment with line 218.
  • The dimension of tested joints should be marked.
  • In which places the cracks have been ddetected? In offshore structures, the common place is in the HAZ (e.g. https://doi.org/10.1007/s00170-020-05617-y). Have you confirmed this statement? Have you observed cracks in the welds?
  • Fig. 4, 6, 8, and so on... - the dimension (scale bars) are missing.

Finite Element Analysis:

  • Fig. 10 -dimensions are needed.
  • Equations need references.
  • Table 9 - the quality is poor. Please redrawn this table, not insert as image.

Comparision:

  • In my opinion this section should be renamed to "Discussion", in which you should compare your results together. However, the comparision with other scientific papers is needed. The scientific article shoud include scientific discussion. Please mark the advantages from your work versus other scientiests. Also, the ne results should be marked. Now, the novelty of your work is hard to observe.

Conclusions:

  • Please support results with values from tests.

Author Response

The following represent point-by-point answers to the reviewers’ comments. Appropriate revisions are made in the revised manuscript, as explained hereunder. In the revised version of the manuscript, all the revisions are highlighted in green.

Reviewer # 02 Comments:

Dear Authors,

I have reviewed paper "Fatigue Crack Monitoring of T-type Joints in Steel Offshore Oil and Gas Jacket Platform". Presented investigations fulfil the aims and scope of the journal. My suggestions are listed below.

Response: The authors are thankful to the reviewer for the valuable comments. As suggested, in the revised version of the manuscript, all the comments are properly addressed. The following represent, point-by-point answers to the reviewers’ comments.

General remarks:

You have presented 41 references. Only three of them have been published after 2018. In my opinion you should present more of the newly published articles. The science made big step forward last years, and it should be visible in your paper.

Response: As suggested, in the revised version of the manuscript, more newly published articles have been added in the revised version of the manuscript. The number of references are now 93.

Introduction:

  • As you stated, the offshore structures may undergo failures. However, you have not mentioned any about repairs. There are two ways, one, took the structure outside the water, and second, performing welding repairs in the water. Are there differences? I think it is worth to mention about techniques allowing to perform repairs in the water (e.g., additional stitches, temper bead welding, changing the bead sequence, waterproof coating on the surface of filler materials).
  • You mentioned about cracks, which are present in offshore structures. You have focused on the fatigue cracks. However, in offshore structures made from HSLA steels the most important problem is cold cracking. Please add this information.

Answer: As suggested, in the revised version of the manuscript, details related to the techniques to perform the repairs have been added in the introduction section. The details related to the cold cracking are also added in the introduction section of the revised version of the manuscript.

Fatigue Test:

  • Please add information about dimensions of investigated JZ25-1S WHPA jacket.

Answer: As suggested, in the revised version of the manuscript, the information about the dimensions of JZ25-1S WHPA jacket has been added on page 4 of the revised manuscript.

  • Please name the welding method used in your investigations.

Answer: The carbon dioxide gas protection welding method is used for welding the specimens in the current investigation. The details related to the welding method is added on page 8 of the revised manuscript.

  • It is worth to show the chemical composition of Q345B steel.

Answer: As suggested, in the revised version of the manuscript, Table 3 is added to show the chemical composition of Q345B steel.

  • Table 2 - presented range of parameters are very wide. Please commet this table. When you have used low parameters, when higher. Also, the information about heat input [kJ/mm] should be presented. This is more important factor than others. Please calculate and show these values.

Answer: As suggested, in the revised version of the manuscript, the range of parameters given in Table 2 is explained on page 8 of the manuscript. The information about the heat input [kJ/mm] is presented in Table 2 on page 8 of the manuscript.  

  • 2. - have you used any rules from relevant standards? If yes, please mention the number of these standards. If not, please add information why.

Answer: Yes, in section 2.2, rules from relevant standards were used. In the revised version of the manuscript, the numbers of these standards are mentioned on page 9.

  • 3. why there were differences in the quality of tested joints? Please comment.

Answer: As suggested, in the revised version of the manuscript, the differences in the quality of the tested joints are discussed in section 2.3 on page 10 of the revised manuscript.

  • Line 183 - why these values were used? Are they corresponding with real conditions? It should be straightly marked in your text. The same comment with line 218.

Answer: The load value used in the experiment is much larger than the actual operating conditions, with the aim of producing fatigue cracks within a limited experimental time. As suggested, in the revised version of the manuscript, this is discussed on pages 10 & 13.

  • The dimension of tested joints should be marked.

Answer: As suggested, in the revised version of the manuscript, the dimension of the tested joints are marked in Figure 2 on page 6.

  • In which places the cracks have been detected? In offshore structures, the common place is in the HAZ (e.g. https://doi.org/10.1007/s00170-020-05617-y). Have you confirmed this statement?

Answer: Similar to the conclusion given in (https://doi.org/10.1007/s00170-020-05617-y), in the current study, the cracks were also detected in the heat-affected zone. In the revised version of the manuscript, this statement is confirmed and is discussed in section 2.3 on page 10.

  • Have you observed cracks in the welds?

Answer: Yes, in the current study, cracks were observed in the welds. In the revised version of the manuscript, this is discussed in section 2.3.2 and in Table 5 on pages 13 & 14.

  • 4, 6, 8, and so on... - the dimension (scale bars) are missing.

Answer: As suggested, in the revised version of the manuscript, the dimension (scale bars) are added to Figures 8, 10 & 12 on pages 14, 16 & 17 respectively.

Finite Element Analysis:

  • 10 -dimensions are needed.

Answer: As suggested, in the revised version of the manuscript, dimensions are added to Figure 12 on page 17.

Equations need references.

  • Table 9 - the quality is poor. Please redrawn this table, not insert as image.

Answer: Based on the recommendation of Reviewer # 01, Table 9 has been removed from the revised manuscript.

Comparison:

In my opinion this section should be renamed to "Discussion", in which you should compare your results together. However, the comparison with other scientific papers is needed. The scientific article should include scientific discussion. Please mark the advantages from your work versus other scientiests. Also, the ne results should be marked. Now, the novelty of your work is hard to observe.

Answer: As suggested, in the revised version of the manuscript, the “Comparison” section is renamed to “Discussion”. In the revised version of the manuscript, a comparison between our work and other scientific articles and the advantages of our work versus other scientific studies have been included in section 3.5 on page 23 of the revised manuscript.  

Conclusions:

  • Please support results with values from tests.

Answer: As suggested, in the revised version of the manuscript, the results have been supported with values from the tests in the “Conclusion” section on page 23.

Finally, the authors wish to thank the reviewer for his constructive remarks, which are well-taken and implemented to improve the clarity and quality of the manuscript. We thank you for the time you put in reviewing our paper and look forward to meet your expectations.

Reviewer 3 Report

Fatigue Crack Monitoring of T-type Joints in Steel 3 Offshore Oil and Gas Jacket Platform

The application is very nice, at least real knowledge and good results, it is a pity some points are weak:

  • Some yellow new sentences are not clear….perhaps the modification were not made in a proper way.
  • Some works about joints of dissimilar materials are missed, using mechanical joints, see Proc Int mechanical engineers, Part B. I remind that the work was using friction drilling and tapping (threading) achieving good results and even checking galvanic performance. Did you do in your case? Combination of friction drilling and form tapping processes on dissimilar materials for making nutless joints, is one reference that can introduce you some ideas. I saw they did not do FEM.
  • Finite element analysis model: Ok, but the experimental testing is key when someone is proposing a new method. Figure 16 is a comparison, but how many cases di you analyze?. I see the loading frequency of T-type tube joint test was 1 Hz and magnitude was 180 kN…in total: testing time, number of probes….
  • Wylde: the references are very difficult to be found…are they complete, can you reduce at least to those absolutely required?. Figure 14 and 15 are again interesting.
  • Q345…standards for steels change in each country, give some explanation…UNS number or others?
  • JZ25-1S WHPA jacket located in Bohai Bay, China was used in the current study: very useful, congrats!!!
  • The journal is about Sensors: your machine has any modification?
  • Monitoring…why did you use this in title?
  • Experimental Techniques 40 (6), 1555-1565, the SEM society is discussing now the border effects in some testpiece, as it was in this journal. The International society of experimental mechanics (SEM) has proposed in several work in the journal experimental techniques, in which for instance the work is, https://doi.org/10.1007/s40799-016-0134-5, https://doi.org/10.1007/s40799-016-0058-0,  https://doi.org/10.1016/j.matdes.2011.03.049. ASTM E 8M-04, Standard test methods for tension testing of metallic materials, and fatigue can be also being affected by the way to define the testpiece…how did you cut pieces?
  • A better introductions taking into account above points would be welcome.
  • Did you have some picture of the real platform….it will be nice!!

Author Response

The following represent point-by-point answers to the reviewers’ comments. Appropriate revisions are made in the revised manuscript, as explained hereunder. In the revised version of the manuscript, all the revisions are highlighted in Turquoise colour.

Reviewer # 03 Comments:

Fatigue Crack Monitoring of T-type Joints in Steel 3 Offshore Oil and Gas Jacket Platform. The application is very nice, at least real knowledge and good results, it is a pity some points are weak:

Response: The authors are thankful to the reviewer for the valuable comments. As suggested, in the revised version of the manuscript, all the comments are properly addressed. The following represent, point-by-point answers to the reviewers’ comments.

Q1.      Some yellow new sentences are not clear….perhaps the modification were not made in a proper way. Some works about joints of dissimilar materials are missed, using mechanical joints, see Proc Int mechanical engineers, Part B. I remind that the work was using friction drilling and tapping (threading) achieving good results and even checking galvanic performance. Did you do in your case? Combination of friction drilling and form tapping processes on dissimilar materials for making nutless joints, is one reference that can introduce you some ideas. I saw they did not do FEM.

Answer: The authors are thankful to the reviewer for the valuable comment. As suggested, in the revised version of the manuscript, all the comments are properly addressed. The discussion about the joints of dissimilar materials are now provided in the “Introduction” section, on pages 3 & 4 of the revised manuscript. The investigation performed in the mentioned study is not done in our study. The combination of the friction drilling and form tapping processes on dissimilar materials for making nutless joints is also discussed in the “Introduction” section, on pages 3 & 4 of the revised manuscript.  

Q2.      Finite element analysis model: Ok, but the experimental testing is key when someone is proposing a new method. Figure 16 is a comparison, but how many cases di you analyze?. I see the loading frequency of T-type tube joint test was 1 Hz and magnitude was 180 kN…in total: testing time, number of probes….

Answer: The authors are thankful to the reviewer for the valuable comment. As suggested, in the revised version of the manuscript, the various details are now provided in section 2.2 and in Tables 1, 2, 3, 4, 5 & 6 of the revised manuscript.

Q3.      Wylde: the references are very difficult to be found…are they complete, can you reduce at least to those absolutely required? Figure 14 and 15 are again interesting.

Answer: The authors are thankful to the reviewer for the valuable comment. As suggested, in the revised version of the manuscript, the references from “Wylde” are now completed and only those references are selected that are absolutely required. These references are provided on page 28 of the revised manuscript.

Q4.      Q345…standards for steels change in each country, give some explanation…UNS number or others?

Answer: As suggested, in the revised version of the manuscript, Table 3 is added to show the chemical composition of Q345B steel. All around the world there are a number of other steel grades that are equivalent to the Q345B, this is mentioned on page 8 of the revised manuscript.

Q5.      JZ25-1S WHPA jacket located in Bohai Bay, China was used in the current study: very useful, congrats!!!

Answer: The authors are thankful to the reviewer for the appreciation.

Q6.      The journal is about Sensors: your machine has any modification?

Answer: The authors are thankful to the reviewer for the valuable comment. Yes, modification was done in the machine by mounting PZT transducers on specified locations. As shown in Figure 5 on page 10 of the revised manuscript, PTZ-51 ballast ceramic sheet, s15mm, 1MHz frequency, one-sided lead PZT transducers were mounted 1-2 cm below or on the side of the welds on all the specimens.

Q7.      Monitoring…why did you use this in title?

Answer: The authors are thankful to the reviewer for the valuable comment. The word “Monitoring” is used in the title because during the current study the crack length was monitored during the experiments after the application of cyclic loads on the specimens.

Q8.      Experimental Techniques 40 (6), 1555-1565, the SEM society is discussing now the border effects in some test piece, as it was in this journal. The International society of experimental mechanics (SEM) has proposed in several work in the journal experimental techniques, in which for instance the work is, https://doi.org/10.1007/s40799-016-0134-5, https://doi.org/10.1007/s40799-016-0058-0, https://doi.org/10.1016/j.matdes.2011.03.049. ASTM E 8M-04, Standard test methods for tension testing of metallic materials, and fatigue can be also being affected by the way to define the test piece…how did you cut pieces? A better introduction taking into account above points would be welcome.

Answer: The authors are thankful to the reviewer for the valuable comment. As suggested, in the revised version of the manuscript, the discussion related to the mentioned references from the SEM and the standard test methods for tension testing of metallic materials and fatigue by ASTM E 8M-04 are now provided in the “Introduction” section, on pages 3 & 4 of the revised manuscript. In the current study, the wire cutting machine was used for cutting pieces.

Q9.      Did you have some picture of the real platform….it will be nice!!

Answer: The authors are thankful to the reviewer for the valuable comment. As suggested, in the revised version of the manuscript, the real platform is shown in Figure 1b on page 5 of the manuscript.

Finally, the authors wish to thank the reviewer for his constructive remarks, which are well-taken and implemented to improve the clarity and quality of the manuscript. We thank you for the time you put in reviewing our paper and look forward to meet your expectations.

Round 2

Reviewer 2 Report

Dear Authors,

Your efforts are appritiate. The paper has been improved a lot. I recommand it for publishing.

However, during proofreading, please improve the literature. Some positions started by surnamed (e.g., 68), some started from first letter of name (e.g. 69).

Ref. 69 - one author is missing. Should be Tomków, J., Fydrych, D., Rogalski, G., Łabanowski, J.

Ref. 70 Should be, Tomków, J., Fydrych, D., Rogalski, G. not Jacek, T . etc.,

This manuscript is a resubmission of an earlier submission. The following is a list of the peer review reports and author responses from that submission.

Round 1

Reviewer 1 Report

the results of an experimental campaign performed on different welded joints are reported. A finite element analysis of the experimental tests is also provided, showing a quite poor capability of reproducing the observed fatigue response of the joints.

The title of the paper and the analysis of the state of the art focuse on the problem of structural health monitoring of off-shore structures. The title claims the use of piezoelectric sensors to detect fatigue crack initiation and propagation.

Unfortunately, the rest of the paper presents a completely different analysis. The piezoelectric sensors are only mentioned in a short paragraph at page 6, that gives an unclear explanation of their use. The picture in figure 3 does not help in clarifying the sensing system configuration.

A description of the fatigue experiments is then provided and finally the finite elements analysis of the loading conditions in the three joint geometries is described.  The experimental program is questionable: i) the specimens are gripped very near to the weldments; the boundary constraints unavoidably introduce spurious effects at joints; ii) the loading level is very high and produces fatigue failures in the low cycle regime, which is unrealistic for these structural parts; iii) the hysteresis loop and the local plastic deformation is not studied, in this way very important information on the crack initiation and propagation are lost.

The finite element analysis of the test does not take into account the hysteretic behaviour of the material. It only assumes some DNV S—N- curves and tries to estimate the fatigue lives of the joints. No mention is given about the correlation between the numerical results and the outcomes of the piezo-sensors measurements.

The agreement between experimental results is considered good, but errors in the range between 50 and 80% seems to represent a quite poor correlation even in the estimation of the fatigue lives.

The discussion and conclusions are not focused on the use of piezo-sensors for fatigue crack detection.

The paper has several weak point which detrimentally influence the scientific merit of the presented results. Moreover, the paper fails to analyse the potentiality of the piezo-sensors and is therefore outside the principal scope of the journal. For all these reasons the manuscript is unsuitable for publication and should be rejected.

Reviewer 2 Report

The manuscript considers three types of T-type joints used in offshore jacket structures. Lab specimens are prepared and subjected to fatigue tests. The results a quantified in terms of the number of cycles till cracking and crack loaction. These resutls are then compared with FE simulation results. A good qualitative agreement is found. FE analysis is found to underestimate the average fatigue life obtained in experiments, which makes FE analysis a safe tool to assess the fatigue life.

I find the manuscript clearly structured, relatively interesting and supported with with a valuable experimental work. My recommendation is a minor revision, and my remarks are motly of a technical nature.

1) The title emphasizes piezoelectric sensors, while in the mansucript they seem to play a very minor role. I think that their role in the manuscript should be more prominent or the title should be changed.

2) The abstract is loaded with details and relatively hard to read. Please simplify it, distill and clearly emphasize, in direct terms, the novelty factor and the main findings of the manuscript.

3) Please mark the water level in Fig.1

4) sect.2.1: "According to the design drawings of the jacket platform and similar theories..."
What are "similar theories"?

5) sect.2.2: "Anyhow, in such a case, if the destruction is located at some opposite fork of some quiver means of the privilege system, the overall extent of the transmutation at this scale will be significantly low, exerting some effect of destruction on the PZT entrance sign and so insignificant reactivity of PZT. To achieve a victory over this challenge, some extended limit of searching prevalence may be employed so that the impact of destruction on extra quivering means might be mirrored in the PZT impedance sign."
These two sentences are very unclearly formulated and hard to comprehend. Please rewrite and simplify.

6) Please specify directly what type of a PZT transducer is used. Pleaase show on the scheme (e.g., Fig. 2), where it was mounted to the specimens.

7) Please clearly mark on specimen schemes (e.g., Fig.2):
a) supports (mounts to the base) during the experimental fatigue tests
b) points and directions of the testing force
Fig.11 is much further in the manuscript, and the reader has no clue about these details while reading about the experimental results.

8) Table 3: does it make sense to specify the lengths of the cracks up to 1 um accuracy?

9) The average fatigue lifes (in loading cycles) are stated for the three types of specimens as average values. However, the individual values are widely scattered. It would be helpful to state the standard deviation besides the average value.

10) Why the load amplitude for the tube joint (sect.2.3.3) was larger than for the other two types of joints? It was 180 kN vs. 150 kN. It is astonishing, because the much shorter fatigue life (Table 5) might suggest a smaller loading amplitude instead of a larger one.

11) Fig.12: can the three plots be plotted in a single plot, with a legend?

12) Table 9: please list also the standard deviation of the experiemtnal results.

Reviewer 3 Report

In the paper, a piezoelectric sensor is used to monitor the fatigue crack growth in T-type Joints. A Finite Element model is also created.

A piezoelectric sensor is used, but it is not clear if this sensor can be used in practical applications. The paper is not well written and it is difficult to read. The English language should be revised. Moreover, details on the experimental tests and on the simulations are missing. The experimental data are not properly analysed and the scatter of the experimental results is not taken into account. The Finite Element Method is widely used for the analysis of the stress in components and for their design. The innovative aspects of the work carried out in the paper are not clear.

Abstract

  • The Abstract exceeds the maximum number of allowed words (200 words). Please reduce it according to the Guidelines for Authors.

Section 2

  • Page 3, line 101: “JZ25-1S WHPA jacket located in Bohai Bay, P.R China was applied”. What do the Authors mean with this sentence? Please rewrite it.
  • Page 3, line 112-113: “the shape 112 and size of the typical joints mold should be moderate”. What do the Authors mean with moderate? Please clarify it.
  • Section 2.1: is the procedure described for manufacturing the tested specimens the same used for manufacturing the joined component in service? Please clarify it.
  • Page 5, line 149: what do the Authors mean with “lamella laboratory evaluation”?
  • Section 2.2 is not clear. There are many sentences difficult to read. Please rewrite this Section. For example, what is the destruction (“The overall space among the destruction”)? What do the Authors mean with “quiver”? What do the Authors mean with privilege? “To achieve a victory over this challenge” is not scientific.
  • What do the Authors mean with “The test results of the three types of joints are relatively discrete”?
  • Table 3: how were the loads chosen?
  • Page 8, line 210: “Due to the lag and fluctuation of the fatigue crack time observed in the test, the fatigue life has 210 a certain degree of dispersion. In order to reduce the error..”. The fatigue data are generally characterized by a large scatter. This is intrinsic in the fatigue phenomenon and is mainly correlated to the crack nucleation process. Therefore, the experimental scatter should be taken into account and is not an error”.
  • Section 2: as stated by the Authors, cracks due to corrosion can also form. Were the corrosion cracks considered? Was the interaction between the fatigue loads and the corrosive environment taken into account?

Section 3

  • Page 15, lines 339-340: “These curves have been shown to be a proper approximation of fatigue life.” The SN curves model the relation between stress amplitude and the number of cycles to failure. It is not correct to define them as “an approximation of fatigue life”.
  • Page 15, line 341: What is “extreme cycle fatigue”?
  • Figure 12: How were these curves obtained? Experimentally? Through Finite Element Analyses? How was the stress computed? It is not clear from the text.
  • Table 9: the difference is too large to be considered conservative. A 50% difference can be due to errors in the models. Moreover, it is not clear how the experimental fatigue life is computed.

The English language should be revised. Some sentences are not clear and there are many typos or errors. For example:

  • Page 2, line 20: “Studies on the worldwide nonlinear collapse analysis of three-dimensional steel jackets has also been performed in the literature” replace “has” with “have”.
  • Page 2, line 79: “Offshore jacket platform is very imperative in the oil and gas exploitation area”. Replace “very imperative” with other words.
  • Page 8, line 216: replace “prone to shown” with “prone to BE shown.”
  • Page 15, line 341: “The S-N curve is usually best suited for extreme cycle fatigue. Where the strain remains within the 341 elastic range.” Please replace the full stop (before “Where”) with a comma.